# mHealth and COVID-19: A Bibliometric Study

**DOI:** 10.3390/healthcare11081163

**Published:** 2023-04-18

**Authors:** Wai-Ming To, Peter K. C. Lee

**Affiliations:** 1Faculty of Business, Macao Polytechnic University, Macao SAR, China; 2Keele Business School, Keele University, Staffordshire ST5 5AA, UK; k.lee@keele.ac.uk

**Keywords:** mHealth, COVID-19, bibliometric analysis, China

## Abstract

mHealth, i.e., using mobile computing and communication technologies in health care, has played an increasingly important role in the provision of medical care and undertaking self-health monitoring and management in the past two decades. Specifically, it becomes critically important for health care delivery when governments have been forced to impose quarantines and lockdowns during the spikes in COVID-19 cases. Therefore, this research focuses on academic publications including journal articles, reviews, and conference papers on the use of mHealth during the COVID-19 pandemic. Using a keyword search on “mHealth” (or “mobile health”) and “COVID-19” on 7 January 2023 in Scopus, it was found that 1125 documents were officially published between 2020 and 2022. Among these 1125 documents, 1042 documents were journal articles, reviews, and conference papers. Researchers in the US produced 335 articles, followed by UK researchers with 119 articles, and Chinese researchers with 79 articles. Researchers affiliated with Harvard Medical School published the largest number of articles (31), followed by researchers of University College London with 21 articles and Massachusetts General Hospital with 20 articles. Co-occurrence of keywords analysis revealed four clusters, namely “COVID-19, mHealth, mobile applications, and public health”, “adult, adolescent, mental health, and major clinical study”, “human, pandemic, and epidemiology”, and “telemedicine, telehealth, and health care delivery”. Implications of this study are given.

## 1. Introduction

mHealth refers to “the use of mobile computing and communication technologies in health care and public health” [1]. It has been widely studied in the past two decades because health care providers including hospitals, clinics, doctors, pharmacists and nurses and health care recipients including patients, caretakers and general populations have been looking for better ways to communicate and interact with each other remotely and continuously [2,3,4]. Thus, the number of publications on mHealth (or mobile health) has grown rapidly, from less than 100 publications a year in 2006 to over 500 publications in 2012 then to 1200 in 2014, 2295 in 2017, and over 3000 a year since 2019. This data were obtained using a search of [“mHealth” OR “mobile health”] in “Article title, Abstract, Keywords” in Scopus—one of the largest English abstract and citation databases. Specifically, 1125 documents were officially published during the period 2020–2022 when a keyword search using [“mHealth” OR “mobile health”] and “COVID-19” was conducted on 7 January 2023, implying that about one-eighth of mHealth (or mobile health) publications were related to COVID-19 in both 2021 and 2022. Among the 1125 documents, 62 were systematic reviews, such as [5,6,7,8,9], i.e., literature reviews that uses systematic and explicit methods to identify, select, and critically appraise previously published studies with a clearly defined objective or research question [10]. Yet most of these 62 documents focused on a specific issue and selected only a small number of publications in their reviews [5,6,7,8]. Interestingly, there was only one systematic analysis using a bibliometric approach [9]. El-Sherif and Abouzid [9] focused on the use of mHealth apps during the COVID-19 pandemic and identified 550 published articles from 2020 to 1 February 2021 using Scopus. They identified the most productive authors and countries, and co-authorship networks. Additionally, they found a number of mHealth keywords clusters using co-occurrence analysis [9].

As time passes, many countries prepare for an endemic stage of the COVID-19 pandemic [11,12]. Some mHealth apps, such as contact-tracing apps, may no longer be necessary because many countries and people have disabled such apps since late 2021 [13,14]. However, the use of mHealth for monitoring and managing the consequences of COVID-19 may sustain for several years as many publications on mHealth and COVID-19 were officially published in 2022. In fact, the number of publications on mHealth (or mobile health) and COVID-19 had almost doubled at the end of 2022 after El-Sherif and Abouzid’s [9] study was officially published in June 2022. Thus, it is necessary to perform another systematic review using a bibliometric approach on mHealth (or mobile health) and COVID-19. More specifically, this research aims to answer the following questions. RQ1: How many publications on “mHealth or mobile health” and “COVID-19” were officially published during the period 2020–2022? RQ2: Who were the most productive authors and which institutions or organizations were the most productive? RQ3: Which country was the most productive and which funding bodies were the most supporting? RQ4: Which subject areas were the publications categorized into and which source titles published most research findings? RQ5: Which were the most cited publications? RQ6: Which authors and countries collaborated more frequently in undertaking research in this area? RQ7: Which topics (or themes) on mHealth (or mobile health) and COVID-19 emerged during the period 2020–2022? This study’s findings should shed light on mHealth (or mobile health) research for the COVID-19 pandemic. Specifically, they should reveal the top authors, affiliations, and countries and highlight the key themes emerged in the identified documents.

The remainder of the paper is structured as follows. Section 2 presents an analysis of reviews on mHealth and COVID-19 and bibliometric analysis. Section 3 describes the Materials and Methods used in the study. Results and discussion are elucidated in Section 4 and Section 5, respectively. Finally, Section 6 concludes the paper by highlighting the study’s limitation and prospects for future research.

## 2. mHealth, COVID-19, and Bibliometric Analysis

### 2.1. Reviews of mHealth and COVID-19

Among the 1125 documents on mHealth (or mobile health) and COVID-19 published from 2020 to 2022, there were 152 review articles, including 62 systematic reviews. Yet most of the 62 systematic reviews focused on a specific issue, such as the use of a specialized mHealth app to engage with a particular group of patients during COVID-19 and they frequently selected only a small number of publications in their reviews. For example, McGarrigle and Todd [5] focused on the promotion of physical activity in older people using mHealth in the context of COVID-19. They included five reviews in their study and found that interventions delivered via mHealth might be effective in increasing physical activity of older people in the short term. Dauletbaev et al. [6] performed a review on mHealth monitoring of pediatric asthma before and during the COVID-19 pandemic. Their scoping review identified 25 articles reporting the use of synchronous and asynchronous mHealth in pediatric asthma but only one article addressed asthma management, i.e., monitoring physical activity of asthmatic children during COVID-19 [15]. Messner et al. [7] reviewed and evaluated mobile apps for the management of gastrointestinal diseases. They mentioned that many people used gastrointestinal mobile apps during the COVID-19 pandemic. They evaluated the overall quality of such mobile apps (109) using the Mobile Application Rating Scale and found that the overall quality of apps was moderate [7]. Rausachenberg et al. [8] synthesized the literature on digital interventions that might mitigate the negative impact of COVID-19 on public mental health. They identified 83 documents that met their inclusion criteria and suggested that digital interventions should be particularly useful to mitigate mental health impact due to the practices of physical distancing, quarantine, and restrictions on social contacts during the COVID-19 pandemic. The only systematic review using a bibliometric approach was written by El-Sherif and Abouzid [9]. El-Sherif and Abouzid [9] identified that 550 articles were published from 2020 to 1 February 2021 on the use of mHealth apps during the COVID-19 pandemic using Scopus. They reported that most articles were published in JMIR mHealth and uHealth (33 articles), Journal of Medical Internet Research (27 articles), and JMIR Research Protocols (22 articles). The top productive authors were D. Giansanti, G. Samuel, F. Lucivero, and L. Zhang while researchers in the US produced the largest number of publications (143), followed by researchers in the UK (96). Co-occurrence of keywords analysis produced four clusters including telehealth and artificial intelligence, digital contact tracing apps and privacy, mHealth apps and mental health, and mHealth in public health.

### 2.2. Systematic Review and Bibliometric Analysis

Systematic reviews need to be carried out periodically as academic knowledge and publications have grown continuously in the past decades [16,17]. Specifically, the development of a standardized systematic review process was pioneered by medical scientists so that a reliable process to synthesize the outcomes of various clinical trials using the same, similar, or different treatments on a particular medical problem or disease was established [10,17,18,19]. A systematic review normally begins with a clearly defined objective or research question [17,19]. Researchers identify all previous published materials relevant to the objective from academic databases based on the established inclusion and exclusion criteria. The findings of the identified studies are synthesized to come up with insights to address the objective of the review. When quantitative analysis is essential to summarize previous results on a particular research topic, meta-analysis is carried out to determine effect estimates and their variances [20]. According to the latest preferred reporting items for systematic reviews and meta-analysis statement, i.e., PRISMA 2020 and its associated flow diagram [20,21], a systematic review process with or without meta-analysis includes four major phases, namely identification, screening, eligibility, and included. Donthu et al. [22] compared different review approaches, including systematic review, meta-analysis, and bibliometric analysis. They highlighted that a systematic review is appropriate when the review scope is very specific and the size of identified documents is small, i.e., several tens and manageable for content analysis, while meta-analysis can be conducted for homogeneous studies. Donthu et al. [22] indicated that when the scope of a research is broad and the size of identified documents is large, i.e., many hundreds or more than a thousand, a bibliometric analysis is more appropriate because it can summarize large quantities of bibliometric data and identify the intellectual structure and emerging themes of a research field or topic. As our study is aimed at identifying academic publications on mHealth (or mobile health) research for COVID-19 that has been widely researched in the past three years and understanding the bibliometric and intellectual structure of the selected documents, a bibliometric approach was used.

## 3. Materials and Methods

### 3.1. Materials

In the present study, we used Scopus as the source database because Scopus is the largest curated abstract and citation academic database [23]. Scopus was launched by Elsevier in 2004. Initially, it covered around 14,000 journals with 27 million abstract and citation records [24]. It has grown steadily over the last two decades. As of 7 January 2023, Scopus covered over 44,000 source titles including more than 40,000 journals and 500 conference series, over 335,000 books including those from 1851 book series, 87 million records, 1.8 billion cited references, and 17 million author profiles [25]. More importantly, Scopus works closely with bibliometricians to curate its database for bibliometric studies [23]. 

### 3.2. Methods

On 7 January 2023, we performed a keyword search in Scopus using [“mHealth” OR “mobile health”] and “COVID-19” in “Article title, Abstract, Keywords”. In Scopus terms, the search was “(TITLE-ABS-KEY(“mHealth” OR “mobile health”) AND TITLE-ABS-KEY(COVID-19))”. The search resulted in 1170 documents that were published between 2020 and 2022 with 45 articles-in-press. After excluding those articles-in-press, 1125 documents were officially published between 2020 and 2022. Among these 1125 documents, 1042 documents were journal articles, reviews, and conference papers. The 83 excluded documents were notes (24), letters (21), book chapters (19), conference reviews (14), editorials (4), and erratum (1). Each of the identified 1042 documents was screened manually for relevance (i.e., using mHealth or mobile health technologies during the COVID-19 pandemic) in their abstracts and the inclusion of important information, such as title, abstract, author name, year of publication, source title, volume, page numbers, etc. As there was no obvious missing information, all the identified documents with bibliometric information were exported to a CSV file from Scopus. 

A bibliometric analysis includes two categories of analysis, namely performance analysis and science mapping [22,26]. Performance analysis identifies the most productive authors, affiliations and countries, and the most supportive funding bodies. It also highlights the number of documents categorized into different subject areas, the top sources titles, and the most cited documents. Performance analysis was carried out using Scopus built-in bibliometric functions in the present study. Science mapping reveals the relational aspects of the identified documents. It characterizes the collaborations between authors and between countries, and the scientific relationship between research themes. The CSV file exported from Scopus was analyzed using VOSviewer verson1.6.11 [27,28]. Figure 1 shows the flow diagram of this study. 

## 4. Results

### 4.1. Performance Analysis Using Scopus Tools

Among the selected 1042 documents (719 journal articles, 152 reviews, and 171 conference papers), there were 164 documents published in 2020, 449 in 2021, and 429 in 2022. This analysis gave the answer for RQ1. Scopus shows that D. Giansanti of Istituto Superiore di Sanità (ISS) of Italy was the most productive author with eight documents (six articles and two reviews in journals). The second most productive authors were J. Torous of Harvard Medical School, A. Islam of Daffodil International University, and a group of researchers in the University of Limerick including J. Buckley, A. Razzaq, K. Rekanar, I. Richardson, C. Storni, D. Tsvyatkova, and T. Welsh. Each of them published six documents on mHealth (or mobile health) and COVID-19 during the period 2020–2022. 

Table 1 presents the number of publications by affiliation. There were 31 publications of authors affiliated with Harvard Medical School, 21 publications authors affiliated with University College London, and 20 publications affiliated with Massachusetts General Hospital. Among the top 13 most productive institutions, 7 of them were US universities and medical schools/hospitals and three of them were UK higher education institutions. Thus, RQ2 was answered.

Table 2 presents the number of publications by country. As expected, researchers in the US were in the leading position to explore the use of mHealth (or mobile health) for COVID-19 with a total of 335 publications, followed by UK researchers with 119 publications. China and Australia ranked third and fourth with 79 and 70 publications, respectively. Canada and Germany ranked fifth and sixth with 66 and 65 publications, respectively.

Table 3 presents the top ten funding sponsors. It was found that the top three funding sponsors were the US National Institutes of Health and its centers including the National Center for Advancing Translational Sciences and the National Institute of Mental Health. The fourth most active funding sponsors were Canadian Institutes of Health Research and the Horizon 2020 Framework Programme in the European Union. Thus, RQ3 was answered.

Table 4 presents the number of publications by subject area, listed by Scopus. As expected, most publications (727) were categorized into the area of medicine, followed by computer science with 253 publications. Additionally, 172 documents were categorized into the area of engineering and 97 documents were categorized into the area of health professions.

Table 5 presents the top ten source titles. The top four source titles were published by JMIR Publications. They were JMIR Formative Research with 55 publications, Journal of Medical Internet Research with 52 publications, JMIR mHealth and uHealth with 50 publications, and JMIR Research Protocols with 38 publications. International Journal of Environmental Research and Public Health (by MDPI) ranked the fifth with 35 publications. Therefore, RQ4 was addressed.

Table 6 shows the top ten highly cited publications. Polsinelli et al.’s [29] article was the most highly cited publication with 573 citations. This journal article presented a light convolutional neural network (CNN) approach for detecting COVID-19 from computer tomography images of chests. Polsinelli et al. [29] reported that the accuracy of light CNN in the detection of COVID-19 was over 85% and the average classification time was 7.81 s using a medium-end laptop without GPU acceleration. This accurate and efficient diagnostic tool would be a valuable mHealth (i.e., mobile health technological) tool in developed and developing countries during the COVID-19 pandemic as suggested by Polsinelli et al. [29]. Ahmed et al.’s [30] article was the second most highly cited publication with 337 citations. It was a review article evaluating contact tracing apps based on their key attributes, such as system architecture, proximity estimation, security, privacy, and data management. The third highly cited publication was written by Altmann et al. [31]. It was a journal article attracting 187 citations. Altmann et al. [31] explored the acceptability of app-based contact tracing for COVID-19 using 5995 responses from a largescale, multi-country survey carried out in the US, the UK, Italy, France and Germany. They reported that people’s willingness to install the app was very high even though some people were concerned about the privacy and cybersecurity of contact tracing apps. The fourth highly cited publication was written by Badawy and Radovic [32]. It was a review article attracting 121 citations. Badawy and Radovic [32] highlighted that the delivery of pediatric patient care through telemedicine and virtual health should be accelerated, integrated and streamlined due to the COVID-19 pandemic. They also listed a number of issues, such as cost-effectiveness, the quality of patient care, system readiness, and regulatory changes that need to be addressed in future research. The fifth highly cited publication was written by Liu et al. [33]. It was a journal article attracting 117 citations. Liu et al. [33] presented a novel nanozyme chemiluminescence paper test for rapid detection of COVID-19 antigen. They reported that the test could be completed within 16 min, much shorter than the typical 1–2 h required for a nucleic acid COVID-19 test. Signal detection was feasible using a smartphone camera, enabling the wide deployment of the COVID-19 test through mHealth. The sixth to tenth highly cited publications covered mental health apps, wearable technology, electronic immunization registry data, digital psychiatry, and the use of the Internet of Things in healthcare and physical distance monitoring [34,35,36,37,38]. They attracted 90 to 101 citations. Thus, RQ5 was answered.

### 4.2. Science Mapping Using VOSviewer

Co-authorship analysis could be conducted at author or country level in VOSviewer. When co-authorship analysis was performed based on authors with four publications or more, 46 researchers met the threshold. Figure 2 indicates that D. Giansanti was the most productive author with eight documents as expected (purple in color). Nevertheless, the largest cluster, i.e., co-authorship group (red in color) was formed by 18 researchers including J. Buckley, A. Razzaq, K. Rekanar, I. Richardson, C. Storni, D. Tsvyatkova, T. Welsh, etc. Most of them are affiliated with the University of Limerick, Ireland. The second largest group (green in color) was formed by O. Ciani, M. Cucciniello, M, F. Petracca, and R. Tarricone. Ciani, Petracca, and Tarricone are affiliated with the Centre for Research in Health and Social Care Management in Italy while Cucciniello is affiliated with the University of Edinburg in the UK. The third largest group (blue in color) was formed by B.M. Chaudhry, A. Islam, and M.M. Rahman. Islam and Chaudhry are affiliated with the University of Louisiana at Lafayette in the US while Rahman is affiliated with the Bangladesh University of Engineering and Technology. The fourth largest group (yellow in color) was formed by three Chinese researchers, namely Y. Li, H. Zhang, and W. Wang, who are affiliated with the Shenzhen Center for Disease Control and Prevention, the University of Chinese Academy of Sciences, and Wuhan University, respectively.

When co-authorship analysis was performed based on countries with four publications or more, 52 countries met the threshold and formed seven clusters as shown in Figure 3. The first cluster (red in color) included 17 countries and regions, such as China, Saudi Arabia, Singapore, Indonesia, Malaysia, and Bangladesh. The second cluster (green in color) included nine countries, such as Germany, Switzerland, Japan, Sweden, Denmark, and Jordan. The third cluster (blue in color) included eight countries, such as the US, Netherlands, Spain, Israel, Turkey, and Uganda. The fourth cluster (yellow in color) included seven countries, such as the UK, Canada, India, South Africa, Nigeria, and Brazil. The fifth cluster (purple in color) included six countries—Italy, France, Belgium, Slovenia, Austria, and Peru. The sixth cluster (light blue in color) included five countries—Australia, Iran, Vietnam, Poland, and Romania. The seventh cluster (orange in color) included four countries—South Korea, Ireland, Greece and Pakistan. Therefore, RQ6 was addressed.

Co-citation analysis was performed based on the cited references using VOSviewer. It was found that three articles attracted 15 or more co-citations from the selected 1042 documents with 42,050 cited references. The top one was “Digital mental health and COVID-19: using technology today to accelerate the curve on access and quality tomorrow” in JMIR Mental Health by Torous et al. [39]. It attracted 17 co-citation. The second one was “A brief measure for assessing generalized anxiety disorder: the GAD-7” by Spitzer et al. in Archives of Internal Medicine [40] with 16 co-citation. The third one was “Acceptability of app-based contact tracing for COVID-19: cross-country survey study” by Altmann et al. [31] in JMIR mHealth and uHealth with 15 co-citation.

Co-occurrence of keywords analysis was performed based on “all keywords” and the minimum number of occurrences was set to 25 using VOSviewer. Out of the 7025 keywords, 102 were identified. The most frequent keyword was COVID-19 with 727 occurrences, followed by mHealth with 594 occurrences and human with 581 occurrences. Figure 4 shows that four clusters were formed based on co-occurrence of keywords. COVID-19, mHealth, SARS-CoV-2, mobile application, and mobile health were the five core keywords in Cluster 1 (red in color—33 keywords). Female, adult, male, controlled study, and mental health were the five core keywords in Cluster 2 (green in color—28 keywords). Human, humans, pandemic, pandemics, and epidemiology were the five core keywords in Cluster 3 (blue in color; 21 items) and telemedicine, coronavirus disease 2019, telehealth, review, and health care delivery were the five core keywords in Cluster 4 (yellow in color—20 items). Table 7 shows the top ten keywords of each cluster. Therefore, RQ7 was answered.

## 5. Discussion

The results of this research are presented in two part; the first part (Section 4.1) is a bibliometric analysis that directly analyzes the data from Scopus whereas the second part employs basically the same data but employs a network perspective in the analysis using VOSviewer (Section 4.2). Based on the findings of the first part, the first contribution of our results is to facilitate researchers in identifying the leading institutions, scholars, and studies while endeavoring to understand the latest developments in the field of mHealth in order to formulate their research plans. For instance, by reviewing and studying the recent publications of the top three most productive scholars (i.e., D. Giansanti, J. Torous, and A. Islam) and those in the top three most productive institutions (i.e., Harvard Medical School, University College London, and Massachusetts General Hospital [see Table 1]) and the top ten most cited publications as shown in Table 6, one can very efficiently have a very good grasp of the latest knowledge of this field. Second, our results offer insights to aid researchers in identifying the major sponsors for their mHealth research. For instance, results shown in Table 3 indicate that the three most important funding bodies for recent publications on mHealth are National Institutes of Health, National Center for Advancing Translational Sciences and National Institute of Mental Health; they together sponsored over 100 studies. Third, our results help researchers to select and decide journals as the outlets for their manuscripts. For instance, our results shown in Table 5 indicate that the top three journals which accepted the most publications are JMIR Formative Research, Journal of Medical Internet Research, and JMIR mHealth and uHealth. Finally, the results on the countries of the researchers and the subject areas of the publications under review imply that one important and underrepresented avenue of research is likely the use of mHealth in developing counties. Table 2 indicates that, of the top five countries of researchers of recent mHealth publications, only one of them is a developing country (i.e., China) whereas Table 4 indicates that, of five major research areas, only one of them is related to social contexts (i.e., social sciences). Nonetheless, mHealth is particularly relevant and important for developing counties where healthcare infrastructure and resources are often very limited [41].

In regard to the bibliometric analysis using VOSviewer, the results firstly indicate that, based on the clusters developed using co-authorship analysis at the author level (see Figure 1) and country level (see Figure 2), publications on mHealth generally involved researchers from multiple countries or continents. This implies that researchers may explore some more global funding bodies, such as United Nations, World Health Organization (WHO) or some organizations encouraging collaboration on healthcare research in the global or multiregional levels. Indeed, such global funding bodies currently are not the major sponsors among the existing publications (see Table 3). Second, the results on co-citation analysis offer extra insight to aid researchers to understand the seminal works in the current body of knowledge, thereby facilitating their effort in understanding the current development of this research area and formulating better research plans. Finally, the co-occurrence of keywords analysis resulted in four clusters (see Table 7). By examining some of the distinguishing keywords of these four clusters, it can be deduced that Cluster 1 is concerned with applying mHealth in COVID-19 (example keywords: COVID-19, mHealth and mobile applications); Cluster 2 focuses on examining individual-level factors (example keywords: female, adult, middle aged and adolescent); Cluster 3 pertains to pandemic issues (example keywords: pandemic, epidemiology and preventive health services), and Cluster 4 is about telehealth and health care delivery (example keywords: telemedicine, telehealth, health care delivery and health care personnel). These four clusters can be considered as four separate bodies of knowledge. To facilitate the knowledge advancement of mHealth, researchers can attempt to come up with research plans that integrate two or more of these clusters. For instance, integrating Clusters 2, 3, and 4, a study may investigate the application of telemedicine (from Cluster 4) to examine the chronic diseases of the elderly (from Cluster 2) during pandemic periods (from Cluster 3). Indeed, El-Sherif and Abouzid [9] also reviewed the literature of mHealth to identify four clusters using co-occurrence of keywords. However, their analysis covered publications up to 1 February 2022. Additionally, El-Sherif et al. [42] explored the use of digital health during the COVID-19 pandemic. They identified that 468 documents were published from 2019 to December 2021 and co-occurrence of keywords analysis revealed four clusters. Considering the large number of relevant studies published in the remaining months of 2022, the clusters developed by the current research represent a more up-to-date depiction on the latest patterns of studies in the literature.

## 6. Conclusions

This research offers useful contributions to the literature concerning mHealth during the COVID-19 period by helping researchers identify the most productive or important scholars, institutions, funding bodies and journal outlets. Using a keyword search (“mHealth” OR “mobile health”) and “COVID-19” on 7 January 2023 in Scopus, we identified 1042 journal articles, reviews, and conference papers. D. Giansanti of Istituto Superiore di Sanità (ISS) of Italy was found to be the most productive author with six articles and two reviews in journals. Harvard Medical School was found to be the most productive institutions and the US National Institutes of Health and its centers, such as the National Center for Advancing Translational Sciences and the National Institute of Mental Health, were acknowledged by authors in a large number of publications. With respect to outlets, the top four source titles were published by JMIR Publications. Additionally, our results on clusters of the existing publications revealed that researches focus on (i) the application of mHealth in COVID-19, (ii) individual-level factors, (iii) pandemic-related issues, and (iv) telehealth and health care. They provide useful guidance, facilitating researchers’ efforts in future research formulation.

However, this study has a few limitations. First, we only employed publications listed in Scopus. Future research could review publications of multiple databases in order to come up with more representative findings. Second, our review and analyses did not classify publications into different groups according to the methodology (e.g., discussion-oriented, empirical studies, review, etc.). Future research could classify publications into groups based on different methods and then analyze them separately such that more specific insight could be developed for researchers with the corresponding methodology expertise. Finally, our analysis methods and tools are rather traditional, hence future research may explore the use of more advanced methods, such as natural language processing, to examine abstracts or keywords. 

## Figures and Tables

**Figure 1 healthcare-11-01163-f001:**
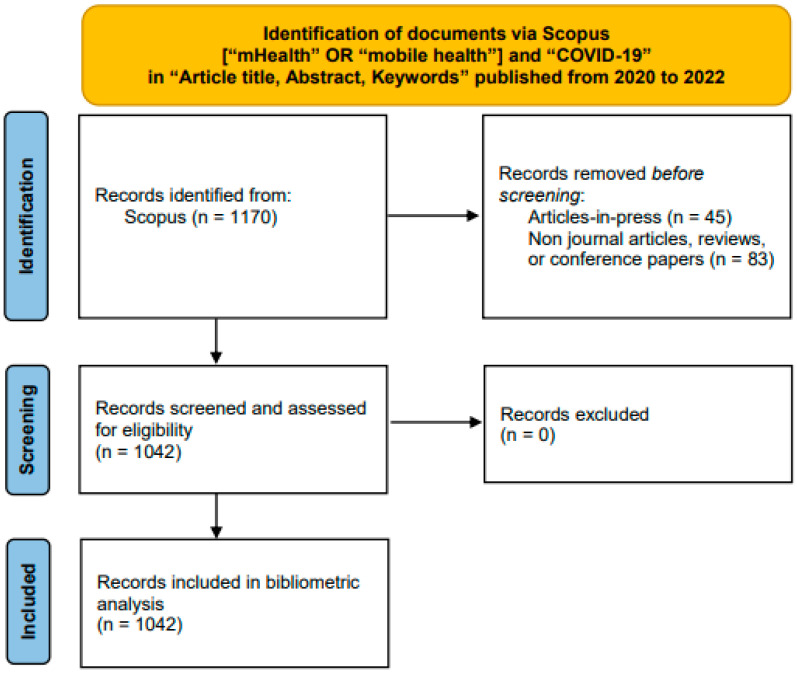
The flow diagram of the study (adapted from PRISMA 2020 [21]).

**Figure 2 healthcare-11-01163-f002:**
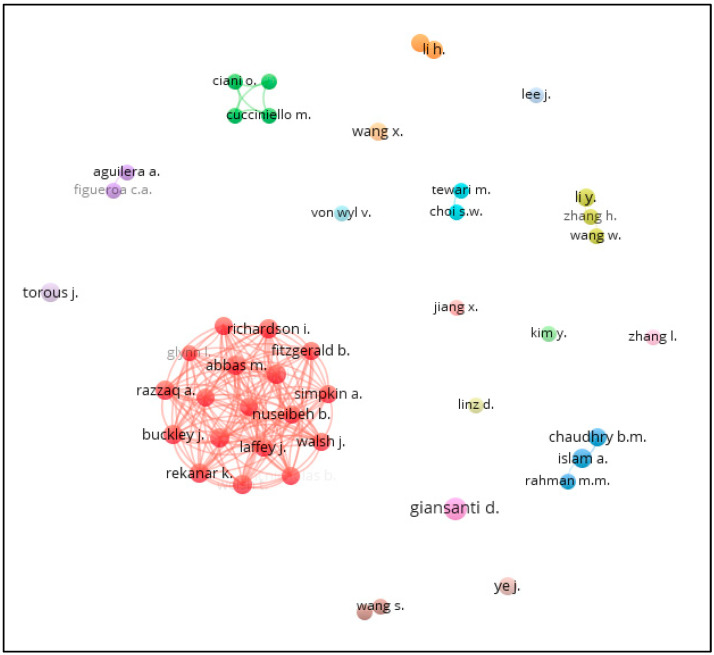
Co-authorship analysis at the author level. (Note: 18 clusters were identified with the largest one (red in color) with 18 researchers; see the text for more details).

**Figure 3 healthcare-11-01163-f003:**
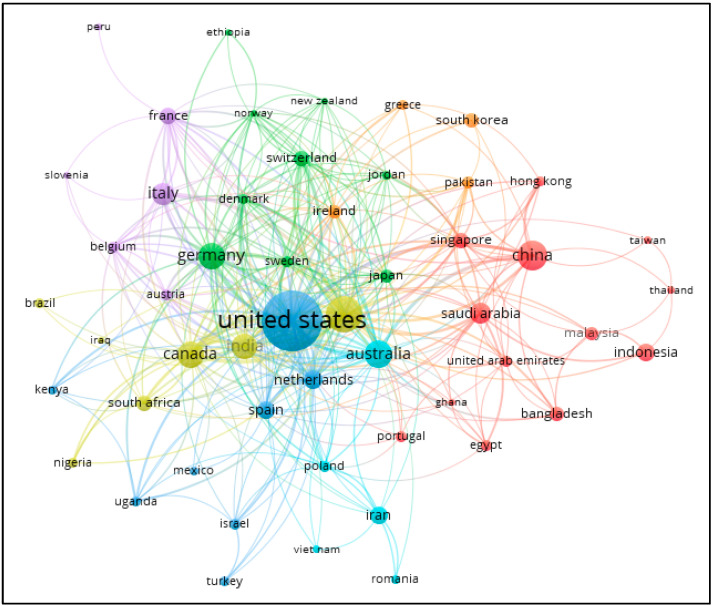
Co-authorship analysis at the country level. (Note: Seven clusters were identified with the largest one (red in color) with 17 countries and regions; see the text for more details).

**Figure 4 healthcare-11-01163-f004:**
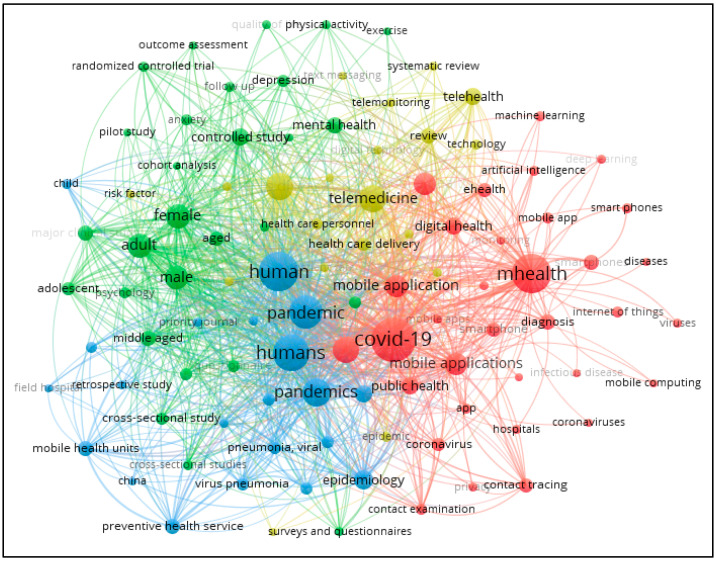
Co-occurrence of keywords analysis. (Note: Four clusters were identified with the largest one (red in color) with 33 keywords; see the text for more details).

**Table 1 healthcare-11-01163-t001:** Top universities/hospitals by the number of publications.

Rank	Affiliation (University/Hospital)	Publications
1	Harvard Medical School	31
2	University College London	21
3	Massachusetts General Hospital	20
4	King’s College London	16
5	University of California, San Francisco	15
6	University of Toronto	14
6	UNSW Sydney	14
8	Johns Hopkins University	12
8	Imperial College London	12
8	University of Washington	12
8	Brigham and Women’s Hospital	12
8	National University of Singapore	12
8	University of Michigan, Ann Arbor	12

**Table 2 healthcare-11-01163-t002:** Top 10 countries by the number of publications.

Rank	Country	Publications
1	United States	335
2	United Kingdom	119
3	China	79
4	Australia	70
5	Canada	66
6	Germany	65
7	India	57
8	Italy	46
9	Saudi Arabia	39
10	Netherlands	34

**Table 3 healthcare-11-01163-t003:** Top 10 funding sponsors.

Rank	Funding Sponsor	Publications
1	National Institutes of Health	75
2	National Center for Advancing Tran. Sci.	18
3	National Institute of Mental Health	18
4	Canadian Institutes of Health Research	15
5	Horizon 2020 Framework Programme	15
6	National Natural Science Foundation of Canada	13
7	Bundesministerium fur Bildung und Forschung	12
8	European Commission	12
9	National Cancer Institute	11
10	National Research Foundation of Korea	10

**Table 4 healthcare-11-01163-t004:** The number of publications by subject area.

Rank	Subject Area	Publications
1	Medicine	717
2	Computer Science	253
3	Engineering	172
4	Health Professions	97
5	Social Sciences	65
6	Decision Sciences	62
7	Nursing	59
8	Biochemistry, Genetics and Mol. Biology	47
9	Environmental Science	45
10	Mathematics	44

Note: A document can be categorized into more than one subject area.

**Table 5 healthcare-11-01163-t005:** Top 10 source titles.

Rank	Source Title	Publications
1	JMIR Formative Research	55
2	Journal of Medical Internet Research	52
3	JMIR mHealth and uHealth	50
4	JMIR Research Protocols	38
5	Int. J. Environ. Res. and Public Health	35
6	Frontiers in Public Health	21
7	Studies in Health Tech. and Informatics	16
8	IEEE Access	13
9	Lecture Notes in Computer Science	12
10	PLOS ONE	12

**Table 6 healthcare-11-01163-t006:** Top 10 highly cited publications.

Author(s)	Title	Year	Source	Citations
Polsinelli et al. [29]	A light CNN for detecting COVID-19 from CT scans of the chest	2020	Pattern Recognition Letters	573
Ahmed et al. [30]	A survey of COVID-19 contact tracing apps	2020	IEEE Access	337
Altmann et al. [31]	Acceptability of app-based contact tracing for COVID-19: cross-country survey study	2020	JMIR mHealth and uHealth	187
Badawy and Radovic [32]	Digital approaches to remote pediatric health care delivery during the COVID-19 pandemic: existing evidence and a call for further research	2020	JMIR Pediatrics and Parenting	121
Liu et al. [33]	Nanoyme chemiluminescence paper test for rapid and sensitive detection of SARS-CoV-2 antigen	2021	Biosensors and Bioelectronics	117
Figueroa and Aguilera [34]	The need for a mental health technology revolution in the COVID-19 pandemic	2020	Frontiers in Psychiatry	101
Ding et al. [35]	Wearable sensing and telehealth technology with potential applications in the Coronavirus pandemic	2021	IEEE Reviews in Biomedical Engineering	99
Chandir et al. [36]	Impact of COVID-19 pandemic response on uptake of routine immunizations in Sindh, Pakistan: an analysis of provincial electronic immunization registry data	2020	Vaccine	95
Torous et al. [37]	The growing field of digital psychiatry: current evidence and the future of apps, social media, chatbots, and virtual reality	2021	World Psychiatry	91
Vedaei et al. [38]	COVID-SAFE: an IoT-based system for automated health monitoring and surveillance in post-pandemic life	2020	IEEE Access	90

**Table 7 healthcare-11-01163-t007:** Top 10 keywords in each cluster with frequency of occurrences.

Cluster 1	Cluster 2	Cluster 3	Cluster 4
COVID-19 (727)	female (229)	human (581)	telemedicine (279)
mHealth (594)	adult (218)	humans (501)	coronavirus disease 2019 (273)
SARS-CoV-2 (269)	male (203)	pandemic (410)	telehealth (120)
mobile application (192)	controlled study (118)	pandemics (309)	review (89)
mobile health (185)	mental health (103)	epidemiology (147)	health care delivery (72)
mobile applications (182)	major clinical study (93)	procedures (108)	health care personnel (52)
digital health (119)	middle aged (91)	mobile health units (95)	epidemic (44)
public health (117)	aged (83)	preventive health service (93)	technology (42)
diagnosis (95)	adolescent (65)	virus pneumonia (62)	digital technology (37)
smartphones (91)	mobile phone (61)	pneumonia, viral (61)	health care (37)

## Data Availability

Data sharing is not applicable to this review.

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
