# Peer review of "mHealth and COVID-19: A Bibliometric Study"

_healthcare, 2023, doi:10.3390/healthcare11081163_

Round 1

Reviewer 1 Report

1. The author should mention if they excluded non-English language publications.

2. The methods section says that each of the 1042 documents were screened manually for relevance. Any specific criteria to determine relevance?

3. The definition of mHealth is pretty loose given that the highest cited article is about the use of convolutional neural network for detecting COVID-19 from CT scan of chests.

Author Response

1st Comment: The author should mention if they excluded non-English language publications.

Our response: Thanks very much for your comment. The study focused on those [“mHealth” OR “mobile health”] and “COVID-19” articles covered by Scopus. Thus, we clarified in the first paragraph of Introduction as “… in Scopus - one of the largest English abstract and citation databases.”

2nd Comment: The methods section says that each of the 1042 documents were screened manually for relevance. Any specific criteria to determine relevance?

Our response: Thanks very much for your comment. In Section 3.2, we clarified that “Each of the identified 1,042 documents was screened manually for relevance (i.e. using mHealth or mobile health technologies during the COVID-19 pandemic) and …”

3rd Comment: The definition of mHealth is pretty loose given that the highest cited article is about the use of convolutional neural network for detecting COVID-19 from CT scan of chests.

Our response: Thanks very much for your comment. We studied this specific paper in detail. In the revised manuscript, we further clarified that “Polsinelli et al.’s [29] article was the most highly cited publication with 573 citations. This journal article presented a light convolutional neural network (CNN) approach for detecting COVID-19 from computer tomography images of chests. Polsinelli et al. [29] reported that the accuracy of light CNN in the detection of COVID-19 was over 85% and the average classification time was 7.81s using a medium-end laptop without GPU acceleration. This accurate and efficient diagnostic tool would be a valuable mHealth (i.e. mobile health technological) tools in developed and developing countries during the COVID-19 pandemic as suggested by Polsinelli et al. [29].”

Dear Reviewer 1: We sincerely hope that we addressed your comments appropriately in the revised manuscript.

Reviewer 2 Report

The topic of this study is important and the paper is overall well-written.

The study is in fact a review-type article, which means that the whole paper is a literature review. Therefore, the second section "2. Literature Review" is somewhat confusing and misleading. The author may reshape the structure of this study for a better understanding of the readers.

Author Response

1st Comment: The topic of this study is important and the paper is overall well-written.

Our response: Thanks so much for your comment.

2nd Comment: The study is in fact a review-type article, which means that the whole paper is a literature review. Therefore, the second section "2. Literature Review" is somewhat confusing and misleading. The author may reshape the structure of this study for a better understanding of the readers.

Our response: Thanks very much for your comment. We reviewed the whole manuscript and renamed the second section as “2. mHealth, COVID-19, and Bibliometric Analysis”.

Dear Reviewer 2: Thanks so much for your time and comments

Reviewer 3 Report

The manuscript is a bibliometric study on mobile health (or mHealth).
The study considers publications on this subject during the period of COVID-19 outbreak and peak infection rates.
The target research publications considered are indeed published during the time frame spanning from 2020 to 2022.
The authors consider a total of 1195 publications from Scopus database and deal with various research questions.

My general opinion of the manuscript is good.
The authors focus onto a relevant domain during an important period.
Also the posed research questions are relevant, although are also typical for a bibliometric study.

I still have the following comments for the authors:
- There are sometimes, throughout the text, sentences repeated.
It is not blind copy-paste, however it makes the reading heavy and difficult.
Examples of this are the research questions or the repeated sentence when referring to publication [9].

-The abstract starts with mHealth without giving a clear definition (just a sentence is enough, like in the Introduction section).

-The Research Questions listed in Sect. I (page 2, lines 61-70) could be better formatted (e.g., "RQ1" etc.) and must be be clearly identified in the next parts of the manuscript, overall when the authors explicitly provide the related answers.

-(page 2, lines 72-76) look to me like a repetition of the introductory part of the research questions.

-Sect. 2.2 is too verbose, can be summarized.

-The authors state (Sect. 3.2): "Among these 1125 documents, 1042 were journal articles, reviews [...]".
What about the 1125-1042=83? Were they posters? Why were they discarded?

-Also, I don't see book chapters anywhere. Were they considered? Were they found?

-The authors state (Sect. 3.2): "Each of the identified 1042 documents was screened manually for relevance [...]".
No information about the manual screening process is given.

-Fig. 1: shows the records identified from Scopus are actually 1170. In the manuscript the authors state only 1125. What about the difference?
In any case, there is no mention about this "1170 documents", so at least the Figure is misleading.

-Sect. 4.1 - line 189: typo -> "Sanita" should be "Sanità" in italian (FYI: the abbreviation of the institute is ISS)

-Sect. 4.1 - line 220: How were the publications categorized? Which field did you consider? Did you consider only the title? The journal classification? What?

Author Response

1st Comment: The manuscript is a bibliometric study on mobile health (or mHealth). The study considers publications on this subject during the period of COVID-19 outbreak and peak infection rates. The target research publications considered are indeed published during the time frame spanning from 2020 to 2022. The authors consider a total of 1195 publications from Scopus database and deal with various research questions. My general opinion of the manuscript is good. The authors focus onto a relevant domain during an important period. Also the posed research questions are relevant, although are also typical for a bibliometric study. I still have the following comments for the authors.

Our response: Thanks so much for your comments. We studied your comments and suggestions thoroughly and revised the manuscript accordingly.

2nd Comment: There are sometimes, throughout the text, sentences repeated. It is not blind copy-paste, however it makes the reading heavy and difficult. Examples of this are the research questions or the repeated sentence when referring to publication [9].

Our response: Thanks for your comment. We rewrote the sentences (for publication [9]) as “…They identified the most productive authors and countries, and co-authorship networks. Additionally, they found a number clusters of mHealth keywords using co-occurrence analysis [9].”

3rd Comment: The abstract starts with mHealth without giving a clear definition (just a sentence is enough, like in the Introduction section).

Our response: Thanks for your comment. As suggested by you, we rewrote the first sentence in Abstract as “mHealth i.e. using mobile computing and communication technologies in health care has played an increasingly important role in the provision of medical care and undertaking self-health monitoring and management in the past two decades.”

4th Comment: The Research Questions listed in Sect. I (page 2, lines 61-70) could be better formatted (e.g., "RQ1" etc.) and must be clearly identified in the next parts of the manuscript, overall when the authors explicitly provide the related answers.

Our response: Thanks for your comment. As suggested by you, we used RQ1, RQ2, ... and RQ7 to label all research questions listed in Section 1. Additionally, answers to these questions were clearly identified in Section 4. For example, in the first paragraph (second sentence), we stated that “This analysis gave the answer for RQ1.” At the end of the second paragraph, we reported that “Thus, RQ2 was answered.”

5th Comment: (page 2, lines 72-76) look to me like a repetition of the introductory part of the research questions.

Our response: Thanks so much for your comment. In the revised manuscript, we rewrote this part as:

“…The study’s findings should shed important light on mHealth (or mobile health) research for the COVID-19 pandemic. Specifically, they should reveal the top authors, affiliations, and countries and highlight the key themes emerged from the identified documents.”

6th Comment: Sect. 2.2 is too verbose, can be summarized.

Our response: Thanks so much for your comment. We rewrote (i.e. shortened) significantly the first half of this section.

7th Comment: The authors state (Sect. 3.2): "Among these 1125 documents, 1042 were journal articles, reviews [...]". What about the 1125-1042=83? Were they posters? Why were they discarded?

Our response: Thanks for your comment. In the revised manuscript, we reported that “The 83 documents excluded were notes (24), letters (21), book chapters (19), conference reviews (14), editorials (4), and erratum (1).” We excluded them because notes, letters, book chapters, conference reviews, editorials, and erratum were not normally considered as “full research articles/reviews” in bibliometric analysis.

8th Comment: Also, I don't see book chapters anywhere. Were they considered? Were they found?

Our response: Thanks for your comment. We did not include book chapters in the current study. Most book chapters would undergo a different review process than journal/conference articles. Additionally, scholars tend to publish their leading-edge research findings in journals/conferences, making a review focusing on journal/conference articles has its unique and important contribution.

9th Comment: The authors state (Sect. 3.2): "Each of the identified 1042 documents was screened manually for relevance [...]". No information about the manual screening process is given.

Our response: Thanks very much for your comment. We clarified that:

“…Each of the identified 1,042 documents was screened manually for relevance (i.e. using mHealth or mobile health technologies during the COVID-19 pandemic) in their abstracts and the inclusion of important information such as title, abstract, author name, year of publication, source title, volume, page numbers, etc. As there was no obvious missing information,…”

10th Comment: Fig. 1: shows the records identified from Scopus are actually 1170. In the manuscript the authors state only 1125. What about the difference?In any case, there is no mention about this "1170 documents", so at least the Figure is misleading.

Our response: The search resulted in 1170 documents. However, 45 of them were “articles-in-press”, resulting in 1125 documents (were officially published). In the revised manuscript, we rewrote Section 3.2 that:

“…The search resulted in 1,170 documents that were published between 2020 and 2022 with 45 articles-in-press. After excluding those articles-in-press, 1,125 documents should be officially published between 2020 and 2022. Among these 1,125 documents, 1,042 documents were journal articles, reviews, and conference papers…”

Thus, Figure 1 is correct.

11th Comment: Sect. 4.1 - line 189: typo -> "Sanita" should be "Sanità" in italian (FYI: the abbreviation of the institute is ISS)

Our response: Thanks so much for your comment. As suggested by you, we corrected them as:

            “D. Giansanti of Istituto Superiore di Sanità (ISS) of Italy…”

12th Comment: Sect. 4.1 - line 220: How were the publications categorized? Which field did you consider? Did you consider only the title? The journal classification? What?

Our response: Each publication was categorized to one or more areas by Scopus (by their in-house experts). Thus, we highlighted it as “Table 4 presents the number of publications by subject area – listed by Scopus.”

Dear Reviewer 3: Thanks so much for your valuable comments that help us improve the manuscript significantly.